# A qualitative review of challenges in recruitment and retention in obstetrics and gynecology in Ireland: The consultants' solution based perspective

Claire M. McCarthy[1]*, Sarah Meaney[2], Suzanne O'Sullivan[1], Mary Horgan[3], Deirdre Bennett[3], Keelin O'Donoghue[4]

1 Cork University Maternity Hospital, Wilton, Cork, Ireland, 2 National Perinatal Epidemiology Centre, Wilton, Cork, Ireland, 3 School of Medicine, University College Cork, Cork, Ireland, 4 The Irish Centre for Maternal and Child Health Research (INFANT), Cork University Maternity Hospital, Wilton, Ireland

* 106408303@ucc.ie

**Data Availability Statement:** Data is held in a repository by one of the authors given its' personal nature. This is a condition of ethical approval for

## Abstract

### Aim

Recruitment and retention remains a concern in obstetrics and gynecology, with consultants having a unique perspective on the daily challenges. We aimed to examine these and examine their solutions to future-proofing the workforce.

### Methods

Primary data were collected from consultant obstetrician-gynecologists in the Republic of Ireland. Using a qualitative methodology, semi-structured interviews were conducted with 17 participants recruited through purposive sampling. Following transcription, deductive content analysis was conducted to identify themes and categories with respect to challenges and solutions in the specialty.

### Results

Findings revealed four superordinate themes of professional and personal factors, opinions of the specialty and the role of the consultant. Respondents expressed fear about low morale in the specialty, but also threats posed by resource availability and training limitations, in addition to medico-legal and media challenges. Solutions centered around re-evaluating training pathways and implementing improved advocacy and support structures for the specialty and for those working within it.

### Conclusions

This study provides a unique standpoint from which to explore an international in obstetrics and gynecology. Its solution-based outlook provides the framework to implement changes to protect and retain the current workforce as well as future-proofing recruitment to secure the specialty.

the study, and is outlined in our revisions letter. A de-identified data set (in the form of interviews with names/locations removed), would still potentially be identifiable when linked with the demographic information of the subjects that is additionally provided. The demographic information provides insight into the career trajectory of respondents and is an essential part of the manuscript. This restriction is placed by the Ethical Committee (contact via crecadmin@ucc.ie). Information could be requested via the Ethical review board, for researchers who meet criteria for access to confidential data.

**Funding:** The author(s) received no specific funding for this work.

**Competing interests:** The authors have declared that no competing interests exist.

# Introduction

Obstetrics and gynecology is a medical specialty that is prone to recruitment and retention issues, which are escalating over time [1]. According to a recent Royal College of Obstetrics and Gynecology report, attrition rates amongst trainees are now approaching 30% [2]. Reasons that have been provided for this include low morale, excessive paperwork, low job satisfaction and a poor work-life balance [3]. Concerted efforts are being made by a number of international representative groups to understand this, in order to reduce and mitigate further attrition [2, 4]. However, any interventions that are made will likely not alter erosion of the specialty for a number of years to come. Therefore, current attrition rates as well as the negative perception of the specialty may continue to have detrimental effects on the numbers entering the specialty, which will ultimately have an impact on care provided to women.

Within medicine, morale among doctors in general has been declining over the past number of years [5]. This decline has been linked with concerns over patient safety and patient care [6]. Obstetrics and gynecology is no different to other medical specialties with its DIT and consultant population being affected by difficulties with work-life balance, fears regarding media scrutiny and litigation as well as concerns with career progression [7–9]. Literature from the last twenty years demonstrates that most doctors experience high levels of work-life imbalance and stress [10]. More recent work also noted that the medico-legal climate and media attention had negative effects on the recruitment and retention of doctors in training (DIT) [8]. The issue of work-life imbalance has also been echoed in studies of more senior hospital doctors, yet this imbalance manifests in different ways, such as an ability to "switch off" and difficulties managing family and professional commitments [11]. Additionally, it has been noted that physicians are fearful of their future, reputation and employment owing to medico-legal claims [12], which can impact on the care that is delivered to patients, both consciously and subconsciously [13].

As specialists in obstetrics and gynecology, consultants can provide insight on the training pathway, the benefits and challenges of their experience, while also have a key role in influencing the direction and continuity of training in the specialty going forward [14]. Through medical training, consultants impart clinical knowledge and skills to junior staff but can also offer advice and mentorship with respect to their career progression, and help them negotiate milestones during their training such as gaining out of program experience and supervising research endeavours [14]. Obstetrics and gynecology consultants who have established careers have witnessed changes in healthcare on the ground, have had opportunities to work internationally in different healthcare systems and have engaged in medical education throughout their working lives [1]. These insights could be integral to the future of the specialty.

While previous work in both obstetrics and gynecology [3, 8, 9] and other medical specialities [11, 15] has focused on the challenges that face medical recruitment and retention, the qualitative solution-based approach to shape a response has not been studied as extensively in the field of obstetrics and gynecology to our knowledge. The World Health Organisation have noted the issue of "medical migration" and some solutions have been created on the basis of financial need, such as mandatory return of service, but these are sometimes without consideration for the clinicians fulfilling this need [16]. A systematic review conducted by Verma et al. has also noted a low quality evidence base to inform recruitment and retention initiatives in the primary care setting, which may be extrapolated to other medical specialties in the absence of similar evidence [17].

We aimed to investigate the challenges faced by consultant obstetrician/gynecologists currently in employment in the Republic of Ireland with respect to their role, the current working climate, the future of the specialty. Additionally, we wanted to explore solutions to improve recruitment and retention within the specialty from the consultant perspective.

## Methods

In choosing the methodology for this study, we reflected that traditional quantitative methodologies would not allow a true and in-depth reflection of the opinions and solutions that our participants would need to express. Thus, a qualitative study was designed in order to fulfil the study objectives.

Purposive sampling was utilized to recruit participants who were working as consultant obstetricians/gynecologists in units around the Republic of Ireland in 2018. An email invitation was extended to all eligible participants who were registered as trainers with the national training body informing them of the scope of the study and nature of the topic. Following distribution of participant information leaflets, all agreed to participate. Written consent forms were completed by all participants prior to the interview being conducted and collected in electronic format for those who completed the interview via telephone, and in paper format for those who completed the interview in person. This was witnessed and counter-signed by the interviewer. All participants were over the age of eighteen. Interviews were conducted at a time and location convenient to each participant, with some taking place over the telephone owing to distance from the base site of the interviewer. Interviews were recorded digitally and transcribed following same.

A semi-structured interview technique was chosen in order to maximize the information obtained from interviewees and capitalize on the wealth of knowledge and experience they could offer. Given that semi-structured interviewing is the most frequent qualitative data source in health services research [18], we felt its use would help inform the design of appropriate interventions. Deductive content analysis was utilized as our analytical medium following our interviews. Content analysis is a research method allowing replicable and valid inferences from data to their context, allowing the provision of new knowledge and facts [19]. It allows researchers to distil words into fewer related categories, and is frequently used in health-related research [20]. Through its application, it can be both descriptive and interpretative which was integral in achieving our aims and objectives [21].

In order to ensure consistency, only one author (SOS) conducted all interviews. An interview guide was designed over several stages by several authors (SOS/DB/MH), and reflected themes arising from the literature nationally and internationally (such as physician morale, [5] impact of the media [8], recruitment and retention [1, 15, 16]). It consisted of 11 main questions (see S1 File). The interview guide focused on three main areas:

i.  Perception of recruitment and retention in obstetrics in gynecology

ii.  Factors impacting morale within the specialty

iii.  Suggested interventions/solutions to help improve the above two factors.

Two pilot interviews were conducted with two consultants (one of whom had a strong qualitative research background) and feedback was incorporated. All participants were known to the interviewing author in their capacity as a practicing consultant colleague and National Specialty Director. This dynamic optimized collaboration and encouraged a more direct and open interviewing style. There was also no need to interpret medical or workplace terminology as the subject matter was intimately known to the interviewer, limiting interruptions to the interviews.

Initial data analysis was conducted by two independent co-authors (CMC/SM). CMC and SM were not involved in the initial construction of the interview guide or conducting the interviews, which ensured there was no undue influence on the interviewees by the interviewer, and analysis of the transcripts were independent. CMC works in the field of obstetrics and

gynecology while SM is a qualitative researcher; this helped to avoid assumptions of shared understandings with interviewees, thus allowing prioritization of the interviewee's voice [22].

Each transcript was analyzed fully with immersion and familiarization of the text by reading and re-reading all transcripts, which identified categories. CMC and SM liaised regularly throughout analysis in order to discuss and refine categories and sub-categories and recognize patterns across the dataset. Recurrent themes were then identified and illustrated with verbatim extracts of texts. Following this, the interviewing author (SOS) was consulted to ensure accurate identification and interpretation of the themes.

Ethical approval was received from the Clinical Research Ethics Committee of the Cork Teaching Hospitals in July 2018 (ECM 4 (kkk) 05/06/18 and ECM 3 (cccccc) 03/07/18.

## Results

Of the seventeen interviews conducted, eleven were held face-to-face and six were conducted by telephone. The mean interview length was 26 minutes (range 17–36 minutes).

The demographics and characteristics of the interviewees are summarized in Table 1. There were nine male consultants and eight female consultants, and eight of the country's nineteen maternity units were represented, including secondary and tertiary maternity units. The mean age of participants was 45.8 years (range 36–60 years). All had spent time training out of program, either though obtaining a higher degree and/or specialist fellowship, with 13 having international experience.

### Challenges in recruitment and retention in obstetrics and gynecology

Four main themes were identified in the analysis of the interview transcripts, as outlined in Table 2. Within these four themes, categories were allocated and classified to the most relevant theme. However, owing to the content, there was considerable crossover between categories. Categories are further expanded with illustrative quotes from the interviews.

**i. Professional factors.** The change in how training is delivered, and its quality was noted by a number of consultants, with many feeling that training was previously more "*onerous*"

**Table 1. Demographic details of participants.**

| Participant | Sex | Age | Years from graduation to Consultancy | Number of children during training | Years spent out of scheme (i.e., fellowship/research) |
|---|---|---|---|---|---|
| 1 | F | 49 | 15 | 3 | 2 |
| 2 | M | 49 | 13 | 4 | 3 |
| 3 | F | 48 | 12 | 2 | 3 |
| 4 | M | 48 | 12 | 2 | 3 |
| 5 | M | 54 | 18 | 2 | 3 |
| 6 | M | 60 | 10 | 1 | 2.5 |
| 7 | M | 40 | 15 | 2 | 3 |
| 8 | F | 50 | 14 | 3 | 3 |
| 9 | M | 60 | 13 | 1 | 4 |
| 10 | F | 39 | 12 | 3 | 2 |
| 11 | F | 46 | 12 | 2 | 2 |
| 12 | F | 42 | 14 | 2 | 1 |
| 13 | F | 36 | 12 | 0 | 3 |
| 14 | M | 54 | 13 | 4 | 4 |
| 15 | M | 60 | 13 | 1 | 4 |
| 16 | M | 48 | 12 | 2 | 3 |
| 17 | F | 43 | 18 | 1 | 3 |

**Table 2. Themes and categories for challenges in recruitment and retention.**

| Themes | Categories |
|---|---|
| **Professional factors** | Training disparities |
| | Lack of resources/inability to provide care |
| | Fear of media/litigation |
| **Personal factors** | Morale |
| | Life challenges |
| | Work culture |
| **Opinion on the specialty** | Job or Vocation |
| | Future of the Specialty |
| | European Working Time Directive |
| | Gender Balance |
| **Role of a Consultant** | Mentor |
| | Trainer |
| | Role Model |

and "*concentrated with steep learning curves*" than current DIT experience. The European Working Time Directive (EWTD) was mentioned a number of times in this context also.

> "*I just wonder if the trainees coming off the scheme after me, because we didn't have the EWTD, are they at the same level in terms of willing to take responsibility, being able to perform the procedures that is required of them*" Consultant 17

The EWTD was felt to negatively affect training, "*it has diluted down the experience*"; however, consultants were largely in support of it.

> "*I feel that the training due to things like the European Working Time Directive, the increased number of trainees, it has diluted down the experience*" Consultant 17

Additional training disparities were noted for European versus non-European trained doctors, and some felt there was on onus on consultants to try to ensure that these doctors excelled and were not disadvantaged.

> "*We need to look at our non-EU doctors. . . we are being really discriminatory*" Consultant 12

The ability to provide a high standard of care within the current resource limitations was also a significant concern, challenging their professional practices in caring for women, but also in providing training to the future of the specialty.

> "*The success of the modest care that we aspire to deliver depends on us having a full complement. . . high-quality fully-trained obstetricians and gynecologists are absolutely central to this*" Consultant 11

> "*We need to think how we deliver care. . . resources are consumed. . .*" Consultant 12

Respondents frequently reflected on the threat perceived by media reporting and litigation, affecting professional and personal lives but also recruitment and retention.

> "*It* [Litigation] *leaves you a bit on edge. . . potentially practicing defensive medicines. . . you are continuously apologizing*" Consultant 10

**ii. Personal factors.** Poor morale was mentioned ubiquitously, having an influence on both personal and professional lives. This was seen to be due to a large number of factors, such as media, litigation, pay parity and patient expectation. Respondents had the feeling of being *"knocked and knocked and knocked"* and morale was described as having reached a *"fatalistic stage"* leading to feelings of hopelessness and despair.

The culture of work and the belief of what it entails was also an important factor that came up in personal considerations of interviewees. There were parallels drawn to work culture as it has evolved over the years, both positively and negatively. Comparisons were made using the phraseology such as *"back in our day"* and the modern *"clock-in, clock-out"* culture, but acknowledgment given that awareness of work-life balance was *"sadly absent in my time"*. These interviews also conveyed a sense of envy of the changes that have since occurred in the culture and working life in obstetrics and gynecology.

**iii. Opinion on the specialty.** There were varied opinions on topics such as the EWTD (as discussed above), gender balance and the future of the specialty. Opinions on gender balance were strongly voiced by both female and male consultants. It was felt that the gender balance of the specialty had heavily shifted in the female direction and this has been a crucial factor in the work-life balance shift; *"people don't want to work as much"*. Interestingly, female consultants were more vocal on this and the language that was used by them was more definitive than their male counterparts.

> *"I think when it's an all-female environment, sometimes, it gets a little bit toxic especially, you know, if people want to go and have a family and their partners cannot move because of their particular job."* Consultant 1

With regard to future expansion of the specialty, there appeared a keen trend to maintain obstetrics and gynecology as a dual specialty in Ireland, with many parallels drawn to the effect of splitting the specialty in the United Kingdom and the perceived negative effect that this has had. It was felt to be an *"absolute retrograde step"*, and it would *"effect the running of everything"*. One respondent believed that the dual specialty had a positive effect on morale:

> *"I like being a good general[ist], . . .that is quite good for the brain and quite good for morale."* Consultant 15

**iv. Role of a consultant.** The multiple aspects of a consultant's role as a peer supporter, trainer and clinician were described by most and they also detailed how this had changed over recent years. There was also acknowledgment of the non-clinical workload that is expected of the modern-day medical practitioner which increases pressure and has an effect on the work-life imbalance.

> *"Doctors do a huge amount outside their contract as regards, you know, guideline development, protocols, contributing to national conversations"* Consultant 8

However, contributing to national issues and advocacy were found to be difficult roles to manage for some respondents, noting that *"you are either* a *professional group or you are advocates politically"*

It was felt that the role of a consultant had changed over the years, nowadays requiring *"way more emotional energy"* and a more hands-on role as a *"provider of care"* rather than leading care, in reference to the move away from the paternalistic nature of medicine in the past.

Given these challenges, some respondents felt that mentorship of more junior colleagues suffered and thus DIT may have a more negative view of consultants and feel unsupported throughout their journey.

*"I am not sure how many of them* [DIT] *feel supported by their consultants, especially in smaller units"* Consultant 5

Again, respondents reflected on their experience, remarking that "*our consultant colleagues had an awful lot more to give in their day*", reflecting the increased difficulties of balancing the challenging modern-day role of the practicing consultant.

The cumulative challenges of their role, and the pressure consultants feel led to descriptions of feelings of despair and frustration.

*"I think it is more and more difficult to be a consultant in today's world. . . much more is expected of you from patients, from society, from the hospital. . . I certainly find it hard to find my way"* Consultant 12

## Solutions for recruitment and retention in obstetrics and gynecology

Following the identification of challenges, the interviewees were then asked to identify solutions that could improve these challenges that were encountered. Again, there was significant overlap in the categories, highlighting a small number of areas where interventions could be conducted within the specialty to improve the challenges encountered. This approach was favored by respondents:

*"If we are going to look for solutions, we have got to look inwards"* Consultant 14

These are outlined in Fig 1, and further expanded on below with illustrative examples from the interviews.

### Re-evaluate training infrastructure

An inclusive and defined training infrastructure was mentioned as a critical solution by most of the interviewees. Early targeted recruitment and highlighting positive aspects of the specialty to those leaving secondary education was mentioned as integral to success.

*"We are asked locally to go to schools when they are doing careers days, it is very important to be involved in that"* Consultant 1

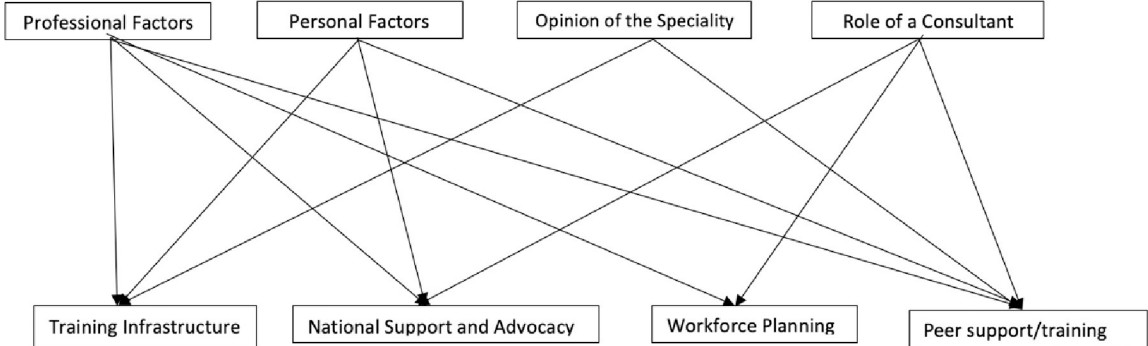

**Fig 1. Themes and categories for solutions in recruitment and retention.**

There were a number of consultants who felt that the training structure in obstetrics and gynecology needed to be re-evaluated and potentially moved to a more competency-based structure with a "*minimum skill set*" to ensure that all individual DIT met these expectations prior to completion of training. The involvement of DIT in the re-design was felt to be integral in this, using anonymous surveys and feedback on different training locations. The link between service provision and how training fits within this, as facilitating a work-life balance, was also considered as essential in moving forward.

> "*we need to get more structured and didactic. . . we have to teach them. . . service and training need to come back in together*" Consultant 17

> "*If we could just have a more fixed kind of location for schemes. . ..no one who wants to work in anything other than Dublin*" Consultant 12

> "*We need to look at alternative pathways really*" Consultant 15

**i. Peer support and training.**   Similarly, the provision of peer support and training was noted to be an imperative adjunct to the role of a Consultant. Mentorship for both DIT and new consultants was mentioned as a tactic to overcome a number of the challenges faced through the working life. The need for a "*positive role model*" and a "*positive mentorship relationship*" for DIT was important and would help in retention.

> "*I personally know of 5 or 6 good trainees who have left obstetrics completely because of an adverse outcome and the lack of support*" Consultant 13

For consultants, the pairing of a "*new consultant with an older consultant*" was also seen as a positive step in peer support, as some noted that "*for* [all] *Irish consultants they don't have any clear supportive structure*".

**ii. Workforce planning.**   It was felt that specific long term workforce planning would improve advocacy for women and those working in the specialty, as well as the quality of DIT exiting the program. This included increasing workforce numbers but also improving resource access. Additionally, it was mentioned that "*maintaining the gender balance*" was essential going forward, and thus attracting more male DIT.

> "*I think better numbers of consultants would help, to support and to help train better our trainees.*" Consultant 5

> "*I worry about trainees as I am not training them. . . if I had appropriate theatre time, I would let them do a lot more work*" Consultant 16

There were also opinions echoing the "*attractive*" role of community gynecology and primary care practices with a special interest in women's health issues, in order allow a choice and expansion in future working options.

> "*So many good people that we have lost, and I think we lose them not just at the end but also throughout their training. . .they just can't see where they will fit in and then we lose them to something else. So, with more flexibility. . .everyone would benefit.*" Consultant 5

**iii. National support and advocacy.**   Given the multiple bodies that are aligned with the specialty, it was felt by respondents that there was a clear role for stronger advocacy for

training and the specialty and also support for the challenges that are faced by staff in the maternity services. It was also felt that it was important to be able to portray a positive outlook of the specialty.

*"Putting out positive messages. . .we need to talk about the good. . . we need to put some balance on the situation"* Consultant 12

Organizations that were mentioned that could be more effective in this support and advocacy role included the training colleges, the National Women and Infants Health Program and also the Institute of Obstetricians and Gynecologists; the latter could *"undertake a real assessment of what future needs are"*.

*"I do think the colleges and, you know, people working do have a responsibility to try and get fairly reasonable information out there to people"* Consultant 8

Comparisons were made to international bodies, where it was felt the support was both more appropriate and collegial.

*"The Royal College of Obstetrics and Gynecology would. . . make statements about issues, they'll involve themselves in the politics and the public and all the rest of it. In Ireland we don't have that. In fact, very often they (Irish institutions) have effectively supported the press against us almost by not being clear and informative to the public"* Consultant 8

Through doing this, it was felt the specialty could have a more positive outlook and this would lead to increased recruitment and retention of DIT in the future.

*"I think we have a huge responsibility unbeknownst to ourselves to talk it up and talk ourselves up"* Consultant 2

## Discussion

This piece of qualitative work discusses the insights of consultant obstetrician/gynecologists working in the maternity and gynecology services in multiple hospitals in the Republic of Ireland. Cumulatively, they have over 228 years of training experience, with all having undertaken additional specialty training and/or research during their training, including international experience. The majority of respondents had children and were working in a variety of secondary and tertiary level maternity hospitals. Many of their opinions and thoughts concurred, giving us insights into the changes and challenges that have faced the specialty over recent years, but those facing the specialty in the years to come. This included reports of increasing workload [5], the challenge of "peri-attrition" [10] and effects of the EWTD [23], which are applicable nationally and internationally. More importantly, consultants also provided valuable insight into solutions that may enable the specialty to work towards reduced attrition and improved recruitment. Overall, improved patient care and satisfaction and clinical performance can be achieved by attracting and recruiting DIT and specialists who allow a culture of supportive safety to prosper in a supportive and welcoming work environment [24, 25].

When we compare the challenges highlighted here to those internationally, we can see that low workplace morale was a common factor [3, 26]. In our analysis, the EWTD was pinpointed as a causative of low morale, however it was also seen to be a positive with regard to improved work-life balance. Despite EWTD having been demonstrated not to affect patient safety and

training [27], many still felt that essential clinical exposure was being neglected by its' implementation [28]. The "shift-work" mentality has also been cited as a "millennial" attitude held by DIT which has been detrimental to the ownership of medical decisions and continuity of care [29].

Challenges surrounding litigation and media scrutiny are well described nationally and internationally in obstetrics [8, 30, 31]. DIT also agree with our group that media scrutiny and litigation fears have a negative effect on recruitment and retention [8]. In order to counteract these fears, support and advocacy initiatives, as well as postgraduate media and legal training would benefit doctors at every level of practice. The fear of litigation has been shown to cause physicians to deviate from routine practice [32], and thus support and provision of this education from training bodies and national programs could provide a greater sense of security to clinicians when facing these challenges; this was also suggested by our interviewees.

Gender was a further issue that was mentioned by both male and female respondents, and despite obstetrics and gynecology being a female-dominated specialty, there was a focus on attracting more males to the specialty and evening out the gender balance. Figures from the United States have shown a change from male to female dominance over the past number of years (from 16% female in 1975 to 83% in the United States) [33, 34], which was referenced to by respondents. However, it has been noted from other research groups that female obstetrician/gynecologists face more significant challenges from a family point of view [35], potentially effecting future work-life balance if changes to working practices are not considered [35]. Gender equity groups are also being promoted internationally in order to create level playing fields to enable men and women to equally realize their potential [36]. Additionally, campaigns such as "Her Time is Now" [34] and programs such as "The 30% Club" [37] are aiming to increase female representation across many medical and non-medical disciplines.

A recurrent solution suggested was in response to frustrations expressed at a lack of response from medical training and representative bodies as well as the government. It was felt that training and representative bodies could assist in providing interpretation and support to healthcare information and strategy. State bodies such as the Department of Health could also place more of an emphasis on womens' health issues and engage with representative bodies to provide a more united approach to healthcare. It was felt that international organizations offered increased support to their members, by routinely releasing position statements and engaging in public health awareness campaigns, whereas the equivalent was not provided nationally. This has been noted in previous literature by Humphries et al., noting ineffectual support from representative bodies in response to political, legal and media pressure [38].

Adequate workforce planning and placing a value on postgraduate training were solutions also focused on by respondents. This has been identified in previous studies [39], and stability in this is known to be beneficial for healthcare workers and healthcare users [40]. An unfavorable postgraduate medical training experience has been blamed for the exodus of DIT [41], and basic specialist trainees have been shown in particular to be more dis-satisfied than those at a higher level [42]. Morgan's opinion piece complements a lot of the findings we have made, commentating on career motivation, attrition and the need for welfare and support in order to mitigate against attrition, but also promote recruitment [1].

This qualitative study has merits in the depth of information provided by respondents and can act as a foundation for improvements nationally and indeed internationally. It complements previously performed quantitative work and offers further information to inform interventions and initiatives to improve recruitment and retention, and thus to secure the workforce of the future.

Consultants working at the forefront of obstetrics and gynecology have provided solutions to a wide range of challenges they face on a daily basis and highlight the need to consider the

evaluation of changes to both recruitment and training practices. National healthcare programs and training bodies should actively engage with consultants to ensure that these solutions can be further developed and then evaluated in practice. While giving consideration to alternative training processes and pathways, an improved consultant training model could also be introduced, providing consultants who are undertaking the delivery of services and training with dedicated and tailored support structures. Through this, the collegial nature of the specialty could be improved, along with changed morale that could positively affect the workforce in the future. A clear national unified voice from organizations representing the specialty and ensuring increased consultant recruitment is also of paramount importance.

This unique perspective from consultants at the forefront of their specialty has revealed the career challenges they encounter, and how they affect the recruitment and retention of DIT who will provide services to the next generation of women and families. A focus through consultant and specialty representative bodies, as well as support from professional colleges and national programs will ensure a workforce that can deliver care into the next generation.

## Supporting information

**S1 File. Interview schedule for qualitative study seeking consultant's insights and solutions to current challenges.**
(DOCX)

**S2 File. COREQ (Consolidated criteria for reporting qualitative research) checklist.**
(PDF)

## Acknowledgments

The authors wish to thank all the participants who gave freely of their time to contribute to our research.

## Author Contributions

**Conceptualization:** Suzanne O'Sullivan, Mary Horgan, Deirdre Bennett, Keelin O'Donoghue.

**Data curation:** Suzanne O'Sullivan.

**Formal analysis:** Claire M. McCarthy, Sarah Meaney, Deirdre Bennett.

**Methodology:** Mary Horgan, Deirdre Bennett.

**Project administration:** Keelin O'Donoghue.

**Supervision:** Mary Horgan, Deirdre Bennett, Keelin O'Donoghue.

**Writing – original draft:** Claire M. McCarthy, Sarah Meaney.

**Writing – review & editing:** Claire M. McCarthy, Suzanne O'Sullivan, Mary Horgan, Deirdre Bennett, Keelin O'Donoghue.

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
