## [Decision Letter · Decision Letter 0]

27 Sep 2022

PONE-D-22-11704A qualitiative review of challenges in recruitment and retention in obstetrics and gynecology in Ireland: the consultants’ solution based perspective.PLOS ONE

Dear Dr. McCarthy,

Thank you for submitting your manuscript to PLOS ONE. After careful consideration, we feel that it has merit but does not fully meet PLOS ONE’s publication criteria as it currently stands. Therefore, we invite you to submit a revised version of the manuscript that addresses the points raised during the review process.

We look forward to receiving your revised manuscript.

Kind regards,

Andrea Bernardes, Ph.D.

Academic Editor

PLOS ONE

2.Please note that in order to use the direct billing option the corresponding author must be affiliated with the chosen institute. Please either amend your manuscript to change the affiliation or corresponding author, or email us at plosone@plos.org with a request to remove this option.

3.In your Data Availability statement, you have not specified where the minimal data set underlying the results described in your manuscript can be found. PLOS defines a study's minimal data set as the underlying data used to reach the conclusions drawn in the manuscript and any additional data required to replicate the reported study findings in their entirety. All PLOS journals require that the minimal data set be made fully available. For more information about our data policy, please see http://journals.plos.org/plosone/s/data-availability.

4.We note that you have indicated that data from this study are available upon request. PLOS only allows data to be available upon request if there are legal or ethical restrictions on sharing data publicly. For more information on unacceptable data access restrictions, please see http://journals.plos.org/plosone/s/data-availability#loc-unacceptable-data-access-restrictions.

a) If there are ethical or legal restrictions on sharing a de-identified data set, please explain them in detail (e.g., data contain potentially sensitive information, data are owned by a third-party organization, etc.) and who has

imposed them (e.g., an ethics committee). Please also provide contact information for a data access committee, ethics committee, or other institutional body to which data requests may be sent.

Additional Editor Comments:

Dear Authors,

Thank you for submitting your manuscript to PLOS ONE. After careful consideration, we feel that it has merit but does not fully meet PLOS ONE’s publication criteria as it currently stands. Therefore, we invite you to submit a revised version of the manuscript that addresses the points raised during the review process.

Reviewers' comments:

Reviewer's Responses to Questions

**Comments to the Author**

1. Is the manuscript technically sound, and do the data support the conclusions?

Reviewer #1: Partly

Reviewer #2: Yes

2. Has the statistical analysis been performed appropriately and rigorously? 

Reviewer #1: N/A

Reviewer #2: N/A

3. Have the authors made all data underlying the findings in their manuscript fully available?

Reviewer #1: No

Reviewer #2: Yes

4. Is the manuscript presented in an intelligible fashion and written in standard English?

Reviewer #1: Yes

Reviewer #2: Yes

5. Review Comments to the Author

Reviewer #1: This manuscript is relevant to the specialty of Obstetrics and gynecology since the authors state the current difficulties on recruitment and retention of doctor in training. Through the qualitative approach of the research, they understood the reasons for the problem and even proposed solutions.

Major points to consider in subsequent versions:

Introduction

The manuscript presents a delimitation of research problem, with a literature review on reasons and impact on specialty. I recommend present relevant controversies about the problem, such as, for example, adding data that demonstrate different realities with positive rates of recruitment and retention.

Method

However, the manuscript has points that meet the guidelines of qualitative review studies, recommend presenting with detail to other researchers can replicate the study.

Therefore, recommend reviewing the manuscript and adhere to qualitative research studies guidelines as described in the submission guidelines - COREQ or SRQR.

Insert details of data collection procedures and analysis: inclusion criteria of the participants; 11 main questions of the interview so that the method is reproducible; researchers' characteristics that may influence the research; start and stop dates of data collection and analysis; methods for processing data prior to and during analysis; techniques to enhance trustworthiness and credibility of data analysis.

Results

The manuscript presents the results organized into relevant themes and their respective categories. I suggest adding definition about The European Working Time Directive (EWTD) and how linked with problem of research, to allow readers outside the field to understand the significance.

Discussion

I recommend explain how the results relation to recent previous studies.

Conclusion

the conclusion this study was described only in the abstract, include in the full text.

Reviewer #2: The aim of the study was to investigate the challenges faced by consultant obstetrician/gynecologists currently in employment in the Republic of Ireland with respect to their role, the current working climate, and the future of the specialty. It was qualitative research, using semi-structured interviews with 17 participants and applying deductive content analysis to identify themes and categories. The results present a unique perspective from consultants at the forefront of their specialty.

The article presents logical chaining and the method was described clearly. For publication, I present the following minor revisions:

(1) In the title, the use “qualitative review” suggests a literature review. I consider better the use of “qualitative view” or “qualitative study”.

(2) In order to further the method, I suggest reviewing the COREQ checklist especially in the itens methodological orientation and theory, non-participation, study design, etc.

(3) To switch the information related to the interview length from Results do Method: “The mean interview length was 26 minutes (range 17-36 minutes)”

(4) Considering the participants were “consultant obstetricians/gynecologists in units around the Republic of Ireland”, I missed the information regarding where they were from. If the authors collect, it would be interesting to detail if the participants were from small cities, capital, rural areas, etc.

6. PLOS authors have the option to publish the peer review history of their article (what does this mean?). If published, this will include your full peer review and any attached files.

Reviewer #1: No

Reviewer #2: No

---

## [Author Response · Author response to Decision Letter 0]

16 Nov 2022

Cork University Maternity Hospital,

 Wilton, 

 Cork, 

 Ireland.

 3 November 2022

Dear colleagues,

Many thanks for the review of our recent article; we appreciate the comments and reviews, and we hope our amendments and responses are welcomed. We have responded to queries in italics below. Lines referenced refer to those in the clean, untracked version submitted.

1. Journal Requirements

Same amended

b. Please either amend your manuscript to change the affiliation or corresponding author

Same amended

c. Data availability statement/PLOS only allows data to be available upon request if there are legal or ethical restrictions on sharing data publicly. 

The minimal data set consists of interviews with a number of practicing Obstetrician/Gynaecologists in the Republic of Ireland. There are multiple instances of identifiable information, as well as statements which could allow patients to be identifiable/scenarios which are utilised to allow examples. From an ethical perspective, the publication of this information is not permitted for this reason. We are able to submit this privately to PLOS ONE, but it is not permissible to publish this data. The Clinical Research Ethics Committee of the Cork Teaching Hospitals is the responsible ethical approval body for this study. 

2. Reviewer 1 

a. The manuscript presents a delimitation of research problem, with a literature review on reasons and impact on specialty. I recommend present relevant controversies about the problem, such as, for example, adding data that demonstrate different realities with positive rates of recruitment and retention.

In our manuscript, as you note, we present the problem of recruitment and retention in our speciality. We have mostly discussed the negative connotations associated with recruitment and retention, as this is largely what our hypothesis and study findings illustrated. However, as per your recommendation, we have included a paragraph on alternative trends and patterns that can be found in the literature:

Inserted at Line 167-171

b. Recommend reviewing the manuscript and adhere to qualitative research studies guidelines as described in the submission guidelines - COREQ or SRQR.

COREQ Checklist utilised and included.

c. Insert details of data collection procedures and analysis: 

i. Inclusion criteria of the participants; those who were “registered as trainers with the national training body”; noted in the methodology of the text

ii. 11 main questions of the interview so that the method is reproducible

Included as Supplementary File 1 

iii. Researchers' characteristics that may influence the research

It is noted that the interviewing author was a working obstetrician/gynaecologist at the time of the interviews, and known to interviewees. In order to limit against bias in reporting, two independent researchers (CMC/SM) conducted the analysis. CMC is an Obstetrician/Gynaecologist and SM is a qualitative researcher. This is noted in the methodology (page 9)

iv. Start and stop dates of data collection and analysis

Added to manuscript: Data was collected between 19th February 2019 and 9th May 2019; data was analysed by CMC/SM in October 2020.

v. Methods for processing data prior to and during analysis

Added to manuscript on page 8/9

vi. Techniques to enhance trustworthiness and credibility of data analysis.

This information is referenced in page 8/9; different authors analysed the data, from two different backgrounds, with one author having a background in qualitative research (and working as a post-doctoral researcher).

d. I suggest adding definition about The European Working Time Directive (EWTD) and how linked with problem of research, to allow readers outside the field to understand the significance.

Explanation and reference to EWTD, including significance; in line 134-138 Research on EWTD included as references.

e. I recommend explain how the results relation to recent previous studies.

Comparisons with previous studies are noted by references 3, 8, 33, 37, 38, 42, 45. Comparisons are included with other research groups findings on workplace morale, challenges with litigation and media scrutiny, work-life balance, poor support from representative bodies. The solutions have also been compared to those in other studies (references 46, 47).

f. The conclusion this study was described only in the abstract, include in the full text.

Conclusion paragraph denoted with heading.

3. Reviewer 2

a. In the title, the use “qualitative review” suggests a literature review. I consider better the use of “qualitative view” or “qualitative study”.

Title changed to “…qualitative study…”

b. In order to further the method, I suggest reviewing the COREQ checklist especially in the items methodological orientation and theory, non-participation, study design, etc

COREQ Checklist utilised and included.

c. To switch the information related to the interview length from Results to Method: “The mean interview length was 26 minutes (range 17-36 minutes)

Moved to Methods section

d. Considering the participants were “consultant obstetricians/gynecologists in units around the Republic of Ireland”, I missed the information regarding where they were from. If the authors collect, it would be interesting to detail if the participants were from small cities, capital, rural areas, etc.

As per the first paragraph in results, the interviewees were from a mix of secondary and tertiary units. Information added to clarify that these would be both urban and rurally located. We have not included this in the Table “Demographic details of participants” as this may lead to clinicians being identifiable and compromise anonymity.

With the inclusion of these changes, we feel that our article provides information on the ongoing topic of recruitment and retention in Obstetrics and Gynaecology, and indeed medicine. Therefore, we appreciate your further reviews, and hope to receive favourable news in the coming weeks. 

We look forward to your review. 

Kind regards,

_________

Dr Claire McCarthy

Corresponding author

---

## [Decision Letter · Decision Letter 1]

12 Dec 2022

A qualitiative study of challenges in recruitment and retention in obstetrics and gynecology in Ireland: the consultants’ solution based perspective.

PONE-D-22-11704R1

Dear Dr. McCarthy,

We’re pleased to inform you that your manuscript has been judged scientifically suitable for publication and will be formally accepted for publication once it meets all outstanding technical requirements.

Kind regards,

Andrea Bernardes, Ph.D.

Academic Editor

PLOS ONE

Additional Editor Comments (optional):

Dear authors,

Thank you for reviewing and improving the paper that made it possible for publication in this Journal.

We hope to receive new submissions in the future.

Regards, Andrea Bernardes

Reviewers' comments:

Reviewer's Responses to Questions

**Comments to the Author**

1. If the authors have adequately addressed your comments raised in a previous round of review and you feel that this manuscript is now acceptable for publication, you may indicate that here to bypass the “Comments to the Author” section, enter your conflict of interest statement in the “Confidential to Editor” section, and submit your "Accept" recommendation.

Reviewer #1: All comments have been addressed

Reviewer #2: All comments have been addressed

2. Is the manuscript technically sound, and do the data support the conclusions?

Reviewer #1: Yes

Reviewer #2: Yes

3. Has the statistical analysis been performed appropriately and rigorously? 

Reviewer #1: N/A

Reviewer #2: N/A

4. Have the authors made all data underlying the findings in their manuscript fully available?

Reviewer #1: Yes

Reviewer #2: Yes

5. Is the manuscript presented in an intelligible fashion and written in standard English?

Reviewer #1: Yes

Reviewer #2: Yes

6. Review Comments to the Author

Reviewer #1: (No Response)

Reviewer #2: (No Response)

7. PLOS authors have the option to publish the peer review history of their article (what does this mean?). If published, this will include your full peer review and any attached files.

Reviewer #1: No

Reviewer #2: **Yes: **José Luís Guedes dos Santos

---

## [Editor Report · Acceptance letter]

16 Dec 2022

PONE-D-22-11704R1 

A qualitative review of challenges in recruitment and retention in obstetrics and gynecology in Ireland: the consultants’ solution based perspective. 

Dear Dr. McCarthy:

I'm pleased to inform you that your manuscript has been deemed suitable for publication in PLOS ONE. Congratulations! Your manuscript is now with our production department. 

Kind regards, 

on behalf of

Dr. Andrea Bernardes 

Academic Editor

PLOS ONE